# The type 3 secretion system requires actin polymerization to open translocon pores

**Brian C. Russo**[1,2¤a]*, **Jeffrey K. Duncan-Lowey**[1¤b], **Poyin Chen**[1,2], **Marcia B. Goldberg**[1,2]*

**1** Center for Bacterial Pathogenesis, Division of Infectious Diseases, Department of Medicine, Massachusetts General Hospital, Boston, Massachusetts, United States of America, **2** Department of Microbiology, Blavatnik Institute, Harvard Medical School, Boston, Massachusetts, United States of America

¤a Current address: Department of Immunology and Microbiology, University of Colorado School of Medicine, Aurora, Colorado, United States of America
¤b Current address: Department of Immunobiology, Yale School of Medicine, New Haven, Connecticut, United States of America
* brian.russo@cuanschutz.edu (BCR); marcia.goldberg@mgh.harvard.edu (MBG)

**Data Availability Statement:** All relevant data are within the manuscript and its Supporting Information files.

**Funding:** This work was supported by NIH grant R01AI081724 to M.B.G., NIH grants F32AI147549

## Abstract

Many bacterial pathogens require a type 3 secretion system (T3SS) to establish a niche. Host contact activates bacterial T3SS assembly of a translocon pore in the host plasma membrane. Following pore formation, the T3SS docks onto the translocon pore. Docking establishes a continuous passage that enables the translocation of virulence proteins, effectors, into the host cytosol. Here we investigate the contribution of actin polymerization to T3SS-mediated translocation. Using the T3SS model organism *Shigella flexneri*, we show that actin polymerization is required for assembling the translocon pore in an open conformation, thereby enabling effector translocation. Opening of the pore channel is associated with a conformational change to the pore, which is dependent upon actin polymerization and a coiled-coil domain in the pore protein IpaC. Analysis of an IpaC mutant that is defective in ruffle formation shows that actin polymerization-dependent pore opening is distinct from the previously described actin polymerization-dependent ruffles that are required for bacterial internalization. Moreover, actin polymerization is not required for other pore functions, including docking or pore protein insertion into the plasma membrane. Thus, activation of the T3SS is a multilayered process in which host signals are sensed by the translocon pore leading to the activation of effector translocation.

## Author summary

The type 3 secretion system (T3SS) is required for the virulence of a variety of bacteria that infect humans. The T3SS forms a pore in the host cell membrane that is a conduit for delivering virulence proteins into the cell. Here, we demonstrate that actin polymerization is necessary to convert T3SS pores into an open conformation that is competent for virulence protein delivery. We find that activation of type 3 secretion proceeds in a multistep process whereby bacteria dock onto the translocon pore and then activate secretion and delivery of virulence proteins.

and T32AI007061 to P.C., and NIH grants T32AI007061, K22AI137296, and F32AI114162 (https://www.nih.gov/), the Massachusetts General Hospital Executive Committee on Research (https://ecor.mgh.harvard.edu/) Tosteson Award, and the Charles A. King Trust Postdoctoral Research Fellowship Program (https://hria.org/tmf/king/), Bank of America, N.A., Co-Trustees, to B.C. R. The funders had no role in study design, data collection and analysis, decision to publish, or preparation of the manuscript.

**Competing interests:** The authors have declared that no competing interests exist.

## Introduction

Type 3 secretion systems (T3SSs) are essential virulence factors of more than 30 gram-negative bacterial pathogens [1,2]. T3SS deliver bacterial virulence proteins—effectors—into the cytosol of a eukaryotic target cells [3–5]. The delivered effectors coopt cellular signaling pathways, which enables the pathogen to establish a replicative niche [6–8].

T3SSs are molecular syringes with a base that spans both membranes of the gram-negative bacterial envelope [9], a needle that is anchored in the base and extends away from the bacterial surface [9,10], and a tip complex that prevents non-specific activation of the system [9,11–14]. Contact of the tip complex with the eukaryotic membrane induces the delivery and insertion of two bacterial proteins into the plasma membrane [11,15] that assemble into a heterooligomeric pore known as the translocon pore [16–20]. The T3SS needle and tip complex stably associate with the translocon pore in a process known as docking [21–23]. Docking establishes a continuous channel from the bacterial cytoplasm to the eukaryotic cytosol and enables the direct delivery of bacterial effectors into the host cytosol. For *S. flexneri*, *Salmonella enterica* serovar Typhimurium, and *Yersinia pseudotuberculosis*, docking depends on the interaction of a cytosolic domain of the translocon pore with host intermediate filaments [21,24] and, at least in *S. flexneri*, a conformational change in the translocon pore induced by the interaction of the pore with intermediate filaments [24] (Fig 1A). In addition, for several pathogens, including *S. flexneri*, *Y. pseudotuberculosis*, and enterohemorrhagic *E. coli*, efficient T3SS effector translocation depends on host actin polymerization [25–31]. Whereas the bacterial proteins that induce actin polymerization are known [30–32], how actin polymerization contributes to effector protein translocation is unclear.

Here, we define the role of actin polymerization in type 3-mediated protein secretion by *S. flexneri*. We find that actin polymerization is required to convert the docking-competent pore to an open and translocation-competent pore. Only in the presence of actin polymerization are bacterial effectors translocated through the pore and delivered into the host cytosol. This conversion to an open pore is associated with actin polymerization-dependent conformational changes in the membrane-embedded pore protein IpaC. The actin polymerization process required for translocon pore opening is distinct from both actin polymerization-dependent membrane ruffling involved in bacterial uptake into cells and the interactions of the translocon pore with intermediate filaments. Together, these data provide mechanistic insight into the role of actin polymerization in the activation of translocon pore function and effector translocation.

## Results

### Actin polymerization is required for type 3 effector protein translocation but not for bacterial docking

To test whether actin polymerization is required for type 3 effector protein translocation, we quantified the delivery of *S. flexneri* effectors into the cytosol of HeLa cells in the presence of the actin polymerization inhibitor cytochalasin D (cytoD) [33]. CytoD treatment significantly reduced the abundance of the effector OspB (detected via a C-terminal FLAG tag, Fig 1B and 1C, $p<0.05$) in the cytosol of HeLa cells. Similarly, the delivery of the effector IpaA (as IpaA-FLAG) was markedly reduced in the presence of cytoD (S1A and S1B Fig, $p<0.05$). The reduced abundance of effector proteins was not attributable to the FLAG tag *per se*, as similar results were obtained with OspB fused to TEM β-lactamase (S1C and S1D Fig, $p<0.001$). These results indicate that inhibition of actin polymerization with cytoD inhibits the delivery of effector proteins into cells.

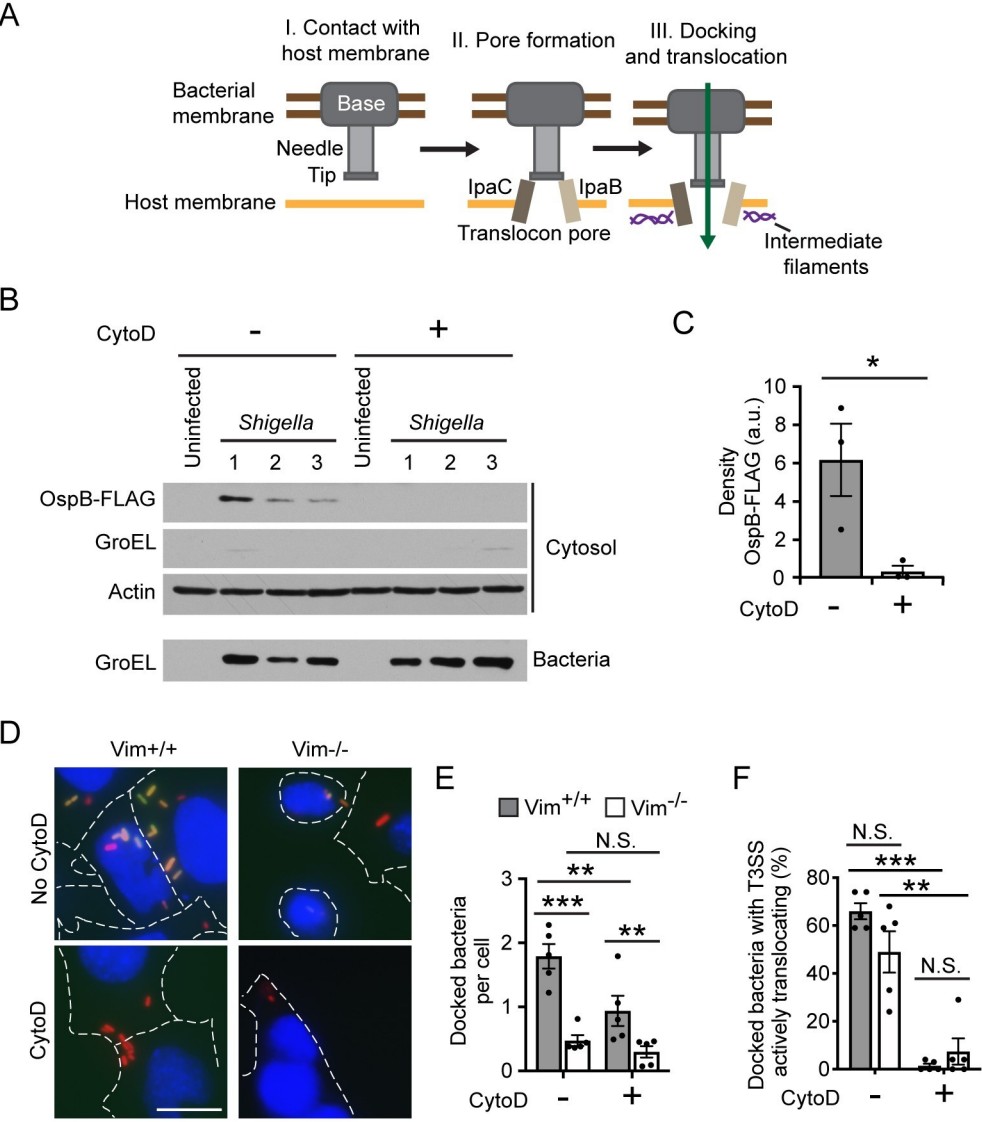

**Fig 1. Type 3 effector translocation requires actin polymerization.** (A) Schematic depiction of stages of type 3 secretion. Transient contact with the host plasma membrane (I) induces the secretion of the translocon pore proteins and their assembly into translocon pores in the host membrane (II). Interaction of the translocon pore protein IpaC with intermediate filaments alters the conformation of the translocon pore (III), which enables bacterial docking and effector protein secretion. (B-C) Effect of CytoD on translocation of FLAG-tagged type 3 effector OspB. *S. flexneri* infection of HeLa cells at a MOI of 200 in the presence or absence of CytoD. (B) Representative western blots. OspB-FLAG, FLAG-tagged effector protein; GroEL, bacterial cytosolic protein; actin, eukaryotic cytosolic protein. All panels are from the same experiment, numbers indicate independent wells from the same experiment. (C) Quantification of FLAG-tagged OspB in experiments represented in panel B. Data points are independent experiments. (D-F) Effect of CytoD on the efficiency of docking and activation of T3SS translocation. *S. flexneri* infection of Vim$^{+/+}$ or Vim$^{-/-}$ MEFs at a MOI of 200 in the presence or absence of CytoD. (D) Representative images at 50 minutes of infection. Blue, Hoechst (DNA); red, mCherry (constitutively produced by bacteria); green, GFP (transcriptionally induced in bacteria by the secretion of the type 3 effector OspD). Dotted white lines, cell boundaries; scale bar, 20 μm. (E) Quantification of docked bacteria per cell from three independent experiments represented in panel D. (F) Percentage of docked bacteria that activate TSAR in experiments represented in panel D. (C) Student's t-test. (E-F) two-way ANOVA with Tukey *post hoc* test. (C, E-F) *, p<0.05; **, p<0.01; ***, p<0.001; data are the mean ± SEM of three to five independent experiments.

When in contact with host cell membranes, the T3SS translocates effectors from the bacterial cytoplasm directly across the eukaryotic plasma membrane into the mammalian cell; effector secretion through the T3SS can be artificially induced from bacteria growing in liquid media in the absence of host cells by the dye Congo red (S1E Fig and [12]). To discriminate whether the effects of cytoD resulted specifically from inhibition of host actin polymerization or from nonspecific effects on T3SS *per se*, we tested the impact of cytoD on Congo red-induced secretion of *S. flexneri* effectors into the extracellular space. Under these conditions, cytoD had no effect on type 3 secretion of effectors (S1F and S1G Fig), demonstrating the effect of cytoD was specific to translocation. Together, these data show that type 3 mediated effector protein translocation requires actin polymerization.

The interaction of intermediate filaments with IpaC induces a conformational change in the pore that is required for efficient docking (Fig 1A, and [21,24]). We therefore investigated whether actin polymerization-induced effector translocation depended on this interaction. Using mouse embryonic fibroblasts (MEF) that encode or do not encode the intermediate filament vimentin (Vim$^{+/+}$ and Vim$^{-/-}$), we compared the number of *S. flexneri* that docked in the presence and absence of cytoD; in these cells, vimentin is the only intermediate filament expressed [34]. Since the number of bacteria that dock to cells will affect the amount of translocated protein, here we assessed translocation efficiency as the percentage of docked bacteria actively translocating effectors by using the fluorescent reporter TSAR (transcription-based secretion activity reporter, S2A Fig and [35]), which reports on T3SS effector secretion in individual bacteria. TSAR is activated when the effector protein OspD is secreted, which liberates the transcription factor MxiE and enables production of GFP from a MxiE-dependent promoter [36]. Consistent with our prior findings [21], *S. flexneri* docked five-fold more efficiently to Vim$^{+/+}$ cells than to Vim$^{-/-}$ cells (Fig 1D and 1E). CytoD reduced bacterial docking to Vim$^{+/+}$ cells two-fold, but had no impact on bacterial docking to Vim$^{-/-}$ cells (Fig 1D and 1E). Among docked bacteria, actin polymerization was significantly required for T3SS effector translocation irrespective of the presence or absence intermediate filaments (Fig 1D and 1F, p<0.001 and p<0.01, respectively). These results indicate that actin-polymerization is required for type 3 effector translocation independently of intermediate filament-induced stabilization of docking and intermediate filament-induced conformational changes in the pore. Moreover, these data show that docking and effector translocation are functionally separable processes.

To test whether the dependence on actin polymerization is generalizable to other cell types, we tested the effect of cytoD on TSAR activation during *S. flexneri* infection of HeLa cells. Consistent with our findings in MEFs, T3SS effector translocation was markedly diminished when actin polymerization was inhibited in HeLa cells (S2B and S2C Fig, p<0.05). To test whether the effect of cytoD is generalizable to other chemicals that inhibit actin polymerization, we tested for an effect of latrunculin B on TSAR activation during *S. flexneri* infection of HeLa cells. Latrunculin B binds to actin monomers and prevents their incorporation into actin filaments. Similar to cytoD, latrunculin B inhibited activation of TSAR (S2D and S2E Fig), which is consistent with previous reports showing latrunculin B blocks effector translocation by *S. flexneri* [30]. The concentrations of cytoD and latrunculin B used in our assays are sufficient to inhibit actin polymerization, as demonstrated by their ability to prevent *S. flexneri* from forming the actin tails used for bacterial motility within the cell cytosol (S3 Fig). Altogether, these data show that type 3 effector translocation is disrupted when actin polymerization is inhibited.

## Actin polymerization is required to form open translocon pore complexes

Since actin polymerization was required for translocation but not docking, we investigated how actin polymerization alters the translocon pore. We employed an assay in which the

formation of translocon pores in the plasma membrane causes release of the fluorescent dye BCECF from BCECF-loaded mammalian cells (Fig 2A). Because dye release is prevented by the presence of the >30 effectors in *S. flexneri* [21,37,38], we used *E. coli* expressing the *S. flexneri* T3SS (*E. coli* pSfT3SS) in the absence of all but four type 3 effectors, as previously described [21,39]. Upon addition of cytoD, the release of dye from cells was reduced from 75% of cells to fewer than 5% of cells (Fig 2B and 2C, p<0.001). Dye release depended on expression of the T3SS, induced in these experiments by IPTG (Fig 2B and 2C), and on bacterial infection, with the amount of release correlated with the MOI used for the infection (S4 Fig and (21)). These results indicate that actin polymerization is required for the formation of an open translocon pore channel.

## Plasma membrane insertion of translocon pore proteins is independent of actin polymerization

We examined the possibility that actin polymerization was required to deliver sufficient pore protein into the plasma membrane by isolating plasma membranes from *S. flexneri* or *E. coli* pSfT3SS infection of cells performed in the presence or absence of cytoD. Actin polymerization had no impact on the efficiency of insertion of translocon pore proteins into membranes by either *S. flexneri* or *E. coli* pSfT3SS (Fig 2D–2F). These results demonstrate that actin polymerization is required for open pore complex formation but does not impact pore protein insertion into the plasma membrane.

## Actin polymerization alters the conformation of the translocon pore

The conformational change induced by the interaction of intermediate filaments with IpaC is associated with altered accessibility of IpaC residues to the extracellular surface [24,40]. To test whether the conformation of the translocon pore is also altered by actin polymerization, we assessed the impact of actin polymerization on the extracellular accessibility of IpaC residues by monitoring the reactivity of single cysteine substitutions in IpaC with the sulfhydryl reactive probe methoxypolyethylene glycol maleimide (PEG5000-maleimide) [24,40], which does not cross the plasma membrane [41] and is too large to pass through the translocon pore [21,24,40,42]. This approach specifically labels cysteine residues in IpaC accessible from the extracellular surface of the eukaryotic cell (Fig 3A). Since native IpaC lacks cysteine residues, it does not react with PEG5000-maleimide [24,40].

Extracellular accessibility of IpaC residues was determined in plasma membranes isolated from *S. flexneri*-infected cells [21,24,40,43]. By testing in parallel the impact of actin polymerization and the impact of IpaC interaction with intermediate filaments on protein conformation, we assessed the relative contribution of each to the extracellular accessibility of specific IpaC residues. The experimental set-up included strains producing single cysteine substitutions in a wild-type (WT) IpaC backbone and strains producing the same single cysteine substitutions in an IpaC backbone that does not interact with intermediate filaments (IpaC R362W, [21,31]). We compared the extracellular accessibility of four IpaC single cysteine substitutions: S17C, which lies in the extracellular domain; A106C, which lies in the transmembrane span; and S349C and A358C, which are located on the cytosolic side of the plasma membrane in a region of IpaC thought to loop into the pore lumen (Fig 3B and [40]).

Actin polymerization and the interaction of IpaC with intermediate filaments had distinct effects on the accessibility of IpaC residues (Fig 3C and 3D). Consistent with our previous findings [24], accessibility of IpaC A106C and IpaC S349C to PEG5000-maleimide was significantly greater in the presence than in the absence of IpaC interaction with intermediate filaments. Of note, in the presence of actin polymerization, IpaC R362W supports the formation

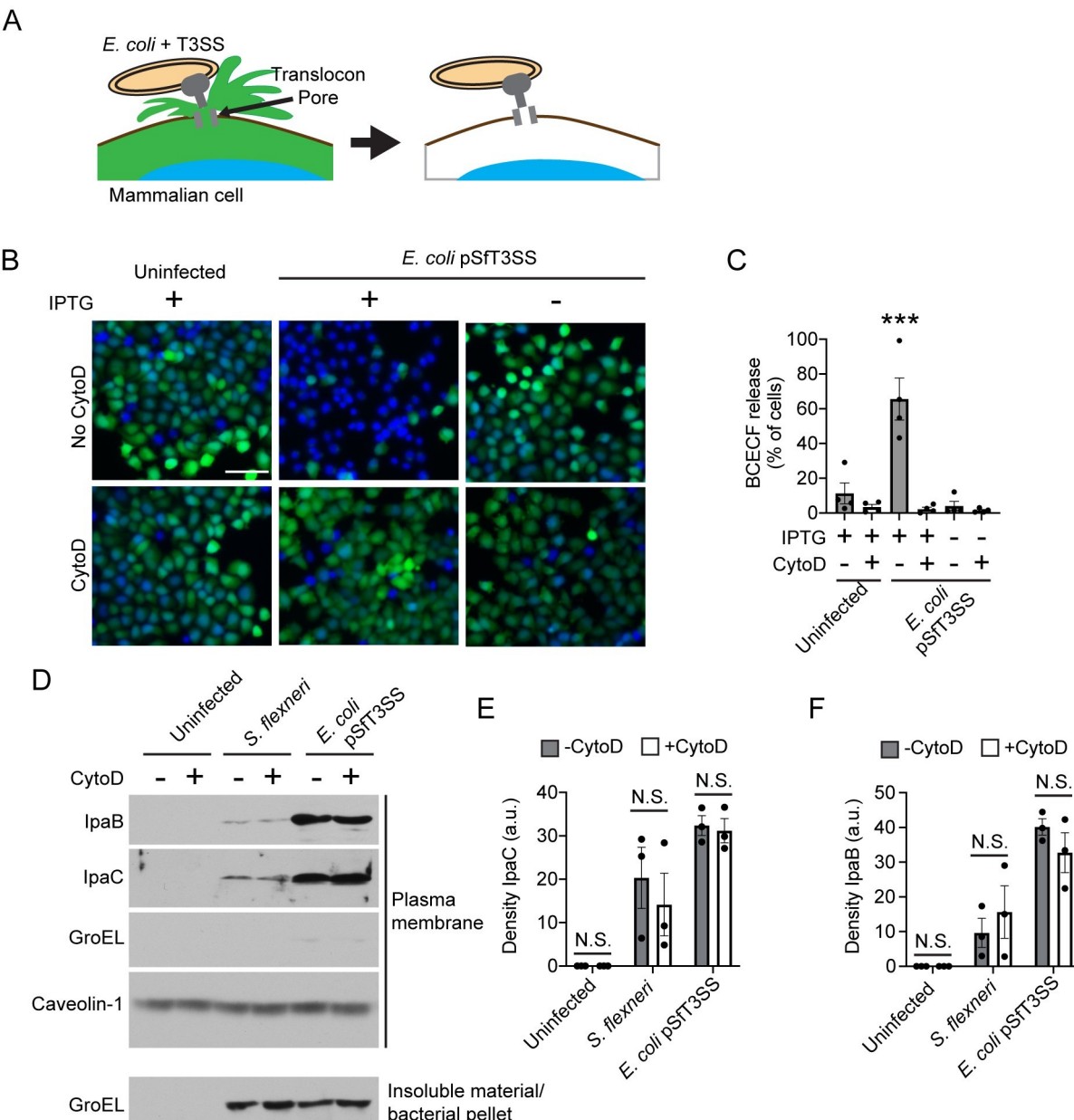

**Fig 2. Actin polymerization is required to form open translocon pore complexes.** (A) Schematic depiction of BCECF dye release from cells that contain a plasma membrane-embedded T3SS translocon pore delivered by *E. coli* producing the *S. flexneri* T3SS (*E. coli* pSfT3SS). (B-C) Effect of CytoD on dye release during *E. coli* pSfT3SS infection at a MOI of 100 of HeLa cells preloaded with BCECF. (B) Representative fluorescent images of cells at 60 minutes of infection. Blue, DNA (Hoechst); green, BCECF; scale bar, 100 μm. (C) Percentage of cells that released BCECF dye. Data points are independent experiments. Data are the mean ± SEM of four independent experiments. ***, p<0.001 by one-way ANOVA with Tukey's *post hoc* test; WT is statistically different from all other conditions. (D-F) The effect of CytoD on the abundance of translocon pore proteins in the plasma membrane during infection of HeLa cells with *S. flexneri* or *E. coli* pSfT3SS at a MOI of 500. (D) Representative western blots. IpaB and IpaC, *S. flexneri* T3SS translocon pore proteins; GroEL, bacterial cytoplasmic protein; caveolin-1, plasma membrane protein. (E-F) Quantification of levels of IpaC (E) or IpaB (F) in the plasma membrane fraction. Data points are independent experiments; data are the mean ± SEM of three independent experiments. N.S., not significant by two-way ANOVA with Sidak *post hoc* test.

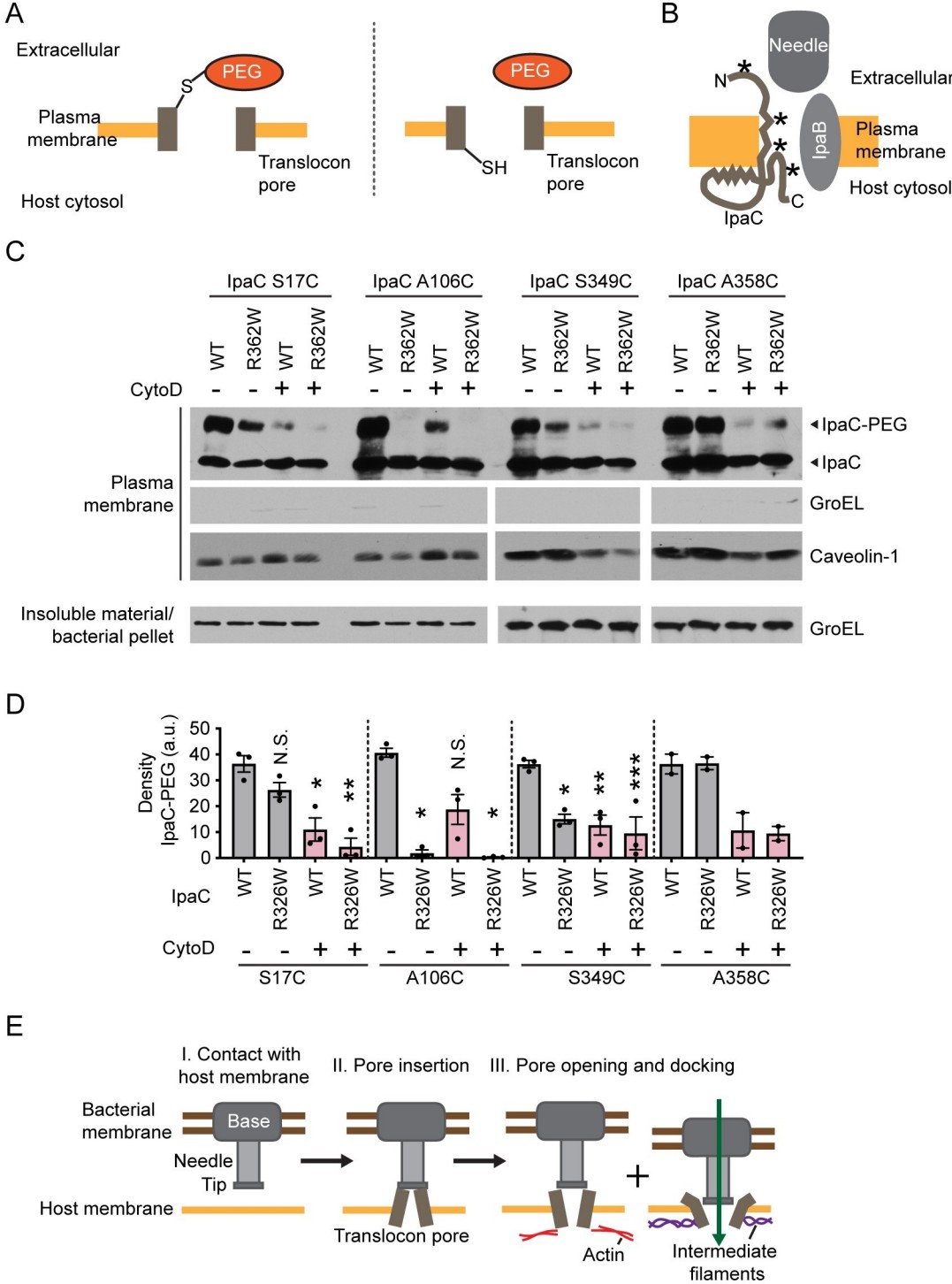

**Fig 3. Actin polymerization induces conformational changes to the translocon pore.** (A) Schematic depiction of extracellular accessibility assay. PEG5000-maleimide (PEG) covalently binds sulfhydryl group of cysteines that are accessible from the outside of the mammalian cell; it is membrane-impermeant and too large to fit through the translocon pore. (B) Schematic of topology of IpaC (from [40]). *, location of cysteine substitutions; top to bottom: S17, A106, S349, A358. (C-D) Effect of CytoD on PEG5000-maleimide labeling of sulfhydryl groups in single cysteine substitution derivatives of WT IpaC or IpaC R362W. *S. flexneri* infection of HeLa cells in the presence or absence of CytoD at a MOI of 500. (C) Representative western blots. WT, WT IpaC backbone; R362W, IpaC R362W backbone; IpaC-PEG, IpaC labeled with PEG5000-maleimide; IpaC, unlabeled IpaC; GroEL, bacterial cytosolic protein; caveolin-1, eukaryotic plasma membrane protein. Data are representative of two to three independent experiments. (D) Quantification of the IpaC-PEG band from experiments represented in panel C. For each IpaC cysteine substitution,

statistical comparisons are relative to accessibility in the WT IpaC backbone background in the absence of CytoD. Data are mean ± SEM of two to three independent experiments. N.S., not significant; *, p<0.05; **, p<0.01; ***, p<0.001. One-way ANOVA with Tukey *post hoc* comparing all means. Statistical analysis was not performed for IpaC A358C because the dataset consisted of only two independent replicates. (E) Model of distinct conformational changes induced by actin polymerization *per se*. Contact with host membrane triggers secretion of pore proteins (I-II), which assemble in the plasma membrane (II). Actin polymerization induces a conformational change associated with opening the translocon pore complexes, and interaction of IpaC with intermediate filaments leads to a conformational change in the pore that enables docking (III). The temporal sequence of the actin polymerization induced conformational changes and the intermediate filament induced conformational changes is uncertain.

of open pores [21]. Accessibility of IpaC A106C to PEG5000-maleimide was only slightly impacted by actin polymerization, whereas in addition to depending on the interaction with intermediate filaments, the accessibility of IpaC S349C to PEG5000-maleimide significantly depended on actin polymerization. In contrast, accessibility of IpaC S17C to PEG5000-maleimide was significantly dependent on actin polymerization, but was independent of the interaction of IpaC with intermediate filaments (Fig 3C and 3D). The impact of actin polymerization on positioning of A106C in the IpaC R362W backbone could not be assessed, as it was inaccessible to PEG5000 labeling both in the presence and absence of cytoD. Thus, the topological positioning of the domains of IpaC evaluated here fall into three categories: IpaC A106, situated in the IpaC transmembrane domain, is shifted markedly by the interaction of IpaC with intermediate filaments. IpaC S17, situated in the extracellular domain, is shifted only by actin polymerization. IpaC S349, situated in a more proximal portion of the cytosolic domain, is shifted by both intermediate filaments and actin polymerization. Together, these data indicate that intermediate filaments and actin polymerization induce distinct conformational (or organizational) states in the translocon pore. Moreover, they show both are required for the generation of a translocation competent pore and for efficient effector protein translocation (Fig 3E).

In parallel, we assessed the role of actin polymerization on the proximity of adjacent IpaC molecules in plasma membrane-embedded pores, by examining the ability of cysteine substitution derivatives of IpaC to crosslink in the presence of the oxidant copper phenanthroline and focusing on derivatives we previously showed are amenable to crosslinking when the pore complex is in a conformation with an open pore channel (24). We tested a cysteine substitution of IpaC at residue A353 (A353C) because this residue is located on the cytosolic side of membrane-embedded IpaC and within the interior of the pore channel (S5A Fig), such that it might provide direct insight into the extent to which the pore channel is open. Since disulfide bonds cannot form in the cytosol, copper-mediated crosslinking at this site should only occur between IpaC monomers present in an open pore complex (S5B Fig). Whereas both IpaC A353C and the extracellular IpaC substitution S17C displayed copper mediated crosslinking in the absence of cytoD, neither showed efficient crosslinking in the presence of cytoD (S5C and S5D Fig, p<0.001 for each). As for cysteine accessibility analyses (above), these oxidative crosslinking data show that actin polymerization induces a distinct conformational (or organizational) state of IpaC that is required to form translocon pores with open pore channels that enable translocation.

## Formation of translocation-competent pores requires the IpaC coiled-coil domain

Within membrane-embedded translocon pores, the coiled-coil domain of IpaC lies on the cytosolic side of the plasma membrane [40]. Because linker scanning mutagenesis of IpaC sequences adjacent to the coiled-coil domain identified residues necessary for *S. flexneri* invasion of cells [44], a process that depends upon T3SS translocation, we tested whether the

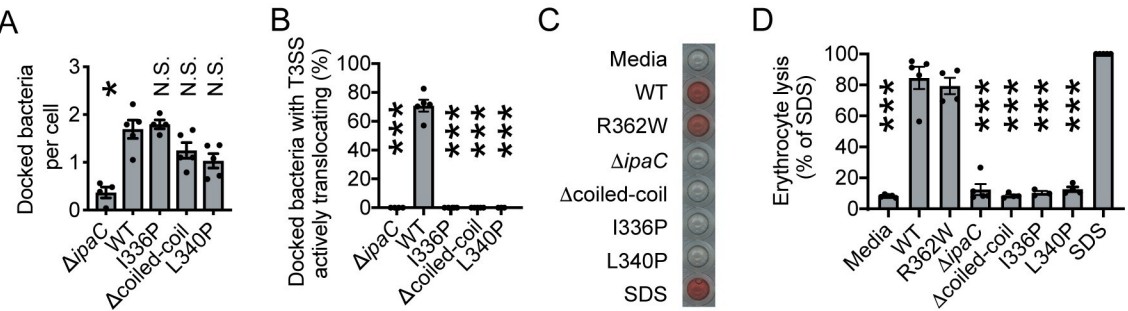

**Fig 4. The coiled-coil domain of IpaC is required to form an open pore but is dispensable for IpaC-mediated docking.** (A-B) Efficiency of docking and effector translocation in infection of MEFs by *S. flexneri* strains producing indicated IpaC protein at a MOI of 200. (A) Docked bacteria per cell. (B) Number of docked bacteria with active secretion. (C-D) Efficiency of pore formation in erythrocytes by indicated *S. flexneri* strains. Infections performed at a MOI of 25. (C) Hemoglobin released from erythrocytes. Representative experiment. (D) Quantification of hemoglobin released in experiments represented in panel C. Data points are individual experiments; data are the mean ± SEM of three to five independent experiments per strain. Statistical comparisons are relative to *S. flexneri* expressing WT IpaC. N.S., not significant; *, p<0.05; ***, p<0.001. One-way ANOVA with Dunnett's *post hoc* test.

coiled-coil domain is required for formation of translocation-competent pores. We compared translocation, docking, and the ability to form open pore complexes by *S. flexneri* producing IpaC that lack the coiled-coil domain (IpaC Δcoiled-coil, residues 308–344, S6A Fig) or producing IpaC containing proline substitutions that disrupt the coiled-coil domain (IpaC I336P and IpaC L340P); prolines prevent oligomerization of coiled-coil domains by inducing a kink in one of the helices [45]. Whereas *S. flexneri* producing IpaC Δcoiled-coil, I336P, or L340P docked to MEFs efficiently, demonstrating that the coiled-coil domain is not required for docking (Fig 4A), they failed to activate the TSAR reporter (Fig 4B, p<0.001 for each), indicating that the coiled-coil domain is required for effector translocation. Consistent with our previous findings [21,24], *S. flexneri* docking was dependent upon IpaC (Fig 4A, p<0.05).

To test whether the defect in translocation was attributable to differences in the ability of these strains to produce open translocon pores, we measured the ability of *S. flexneri* producing IpaC coiled-coil domain mutants to lyse erythrocytes; the formation of open translocon pores in the erythrocyte membrane causes erythrocyte lysis and release of hemoglobin [21,24,40,42]. *S. flexneri* producing IpaC coiled-coil mutants released significantly less hemoglobin than *S. flexneri* producing WT IpaC or an IpaC mutant that is unable to interact with intermediate filaments yet forms pores (IpaC R362W) (Fig 4C and 4D). The inability to form pores is not due to decreased production of these IpaC mutants, as *S. flexneri* strains produce these IpaC alleles at similar levels (S6B Fig). These results indicate that the IpaC coiled-coil domain is dispensable for docking yet is required for the formation of translocon pores with an open channel.

## Actin polymerization-dependent pore opening is distinct from the actin ruffling required for bacterial uptake

Since the coiled-coil domain is required for the formation of an open pore, we sought to identify IpaC residues in the coiled-coil domain required for actin polymerization-dependent pore opening. To identify such residues, we generated libraries of *S. flexneri* IpaC mutants. The first library was generated by replacing charged residues with alanine and expressing them in *S. flexneri* ΔipaC harboring the TSAR reporter, which enabled screening for the impact of the alanine mutation on effector translocation. From this library, we did not identify individual residues required for docking or translocation (S6C–S6E Fig).

As an alternative approach, we generated a library of IpaC mutants of the coiled-coil domain and flanking region using error prone PCR, which does not restrict residue substitutions to alanines. Among 600 *S. flexneri* isolates producing mutants of IpaC, we identified 137 that were significantly impaired compared to bacteria producing WT IpaC in their ability to activate translocation, as determined by the TSAR reporter. Sequencing of *ipaC* alleles from these strains identified 131 mutants with non-sense or frameshift mutations and six mutants with coding substitutions (G296V and S311R, G297V, G297V and S345N, G308P and L309I, A354P, and A354T). We tested strains of *S. flexneri* that express *ipaC* alleles carrying no more than one missense mutation. Whereas TSAR was activated by greater than 80% of bacteria producing WT IpaC, it was activated by only approximately 50% of bacteria producing A354P and approximately 10% of bacteria producing Q308P (Fig 5A–5C, p<0.01 and p<0.001, respectively), indicating that these mutations significantly reduced the efficiency of *S. flexneri* translocation.

We tested whether the failure of these mutants to efficiently translocate was associated with an inability to activate actin polymerization and/or form open translocon pore complexes. As IpaC is required for the actin polymerization-dependent formation of plasma membrane ruffles during bacterial invasion [46], we visualized IpaC nucleation of actin-rich membrane ruffles at sites of bacterial contact with cells. *S. flexneri* producing IpaC A354P or IpaC Q308P were significantly less efficient at inducing the formation of membrane ruffles associated with *S. flexneri* invasion (Fig 5D and 5E). Since translocated effectors contribute to actin polymerization during bacterial invasion, it is possible that the reduced level of translocation observed for bacteria that produce IpaC A354P or IpaC Q308P contributes to the defect in actin ruffle formation, particularly for bacteria producing IpaC Q308P. In contrast, for bacteria producing IpaC A345P, compared to the observed 2-fold reduction in translocation, there is a 4-fold reduction in ruffle formation; this difference suggests that for bacteria producing IpaC A354P, the direct contribution of IpaC to ruffle formation is likely defective.

As actin polymerization is required for the formation of translocon pores with open channels (Fig 2B and 2C), we used the erythrocyte lysis assay to test whether IpaC mutants Q308P and A354P support the formation of pores in erythrocytes. Since these mutants failed to induce actin polymerization-dependent membrane ruffling (Fig 5D and 5E) and supported effector translocation with significantly diminished efficiency compared to WT IpaC (Fig 5A–5C), we anticipated that their ability to support the formation of translocon pores would be defective. Consistent with this hypothesis, pores formed by IpaC Q308P released approximately 50% less hemoglobin than those formed by WT IpaC (Fig 5F and 5G, p<0.001). This mutant supported efficient insertion of the translocon pore proteins into the plasma membrane (Figs 5H and S7), indicating that the defect in hemoglobin release was not due to defects in membrane insertion of translocon proteins and that the pore formed by IpaC Q308P is partially closed. In contrast to our hypothesis, pores formed by IpaC A354P efficiently released hemoglobin from erythrocytes, indicating that this mutant supports formation of a fully open pore channel (Fig 5F and 5G). Interestingly, despite not inducing actin ruffles, bacteria producing IpaC A354P remained sensitive to cytoD treatment, as cytoD inhibited TSAR activation for bacteria producing IpaC A354P, as for bacteria producing WT IpaC (Fig 5I and 5J, p<0.001 for each). These data indicate that the actin dependent process required to form open translocon pores is distinct from actin-dependent ruffle formation.

## Discussion

Here we show that actin polymerization induces conformational changes to the T3SS translocon pore complex that open the channel of the pore and activate effector protein translocation.

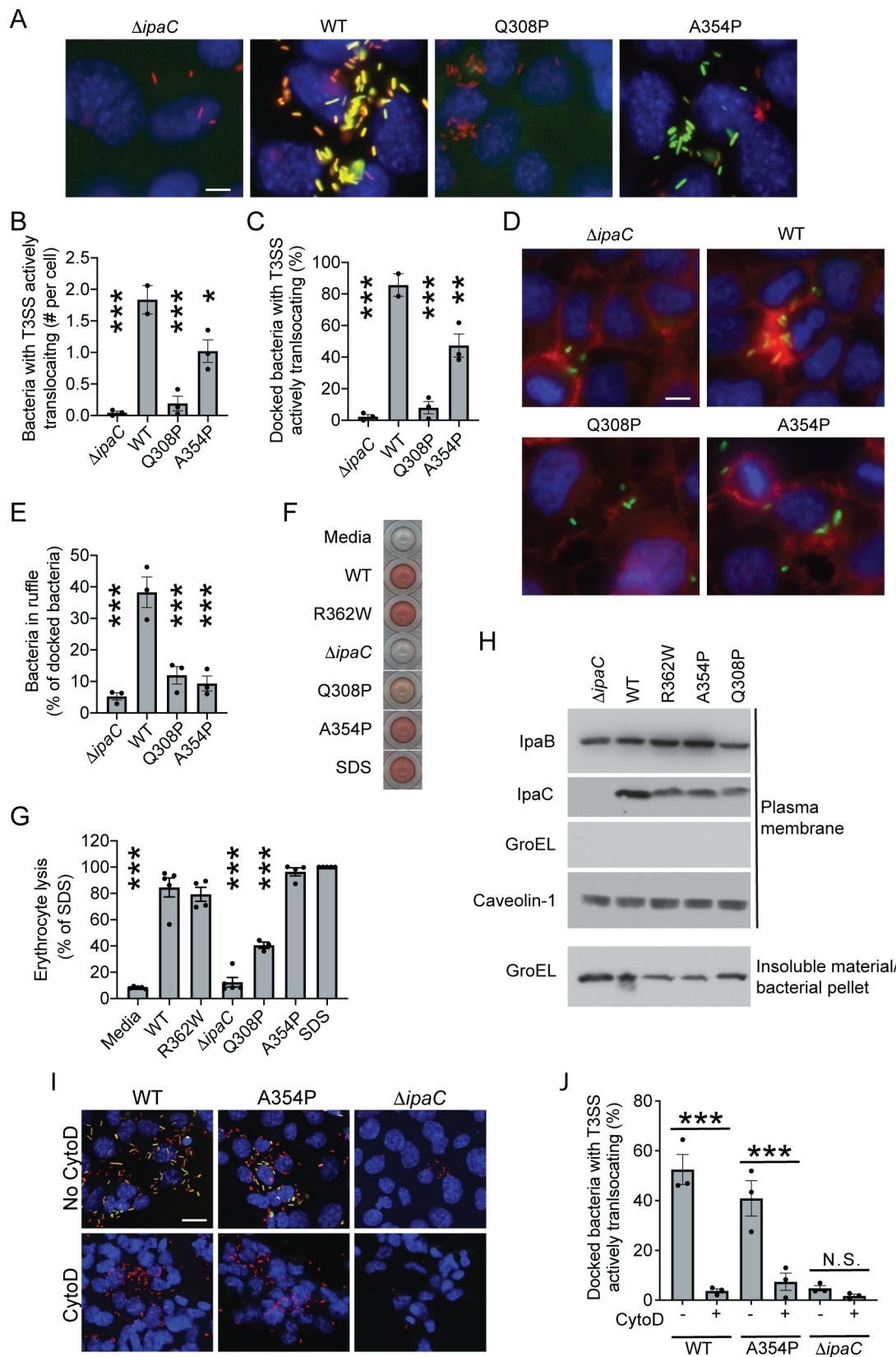

**Fig 5. Opening of the translocon pore is independent of actin polymerization-dependent membrane ruffling.** (A-C) Effect of IpaC mutations on bacterial docking and effector translocation. Infection of MEFs by *S. flexneri* strains producing indicated IpaC protein at MOI of 200. (A) Representative images. Red, mCherry constitutively produced by all bacteria; green, TSAR; blue, DNA (Hoechst). Scale bar, 10 μm. (B) Number of bacteria with active T3SS secretion in experiments represented in panel A. (C) Percentage of docked bacteria with activate T3SS secretion in experiments represented in panel A. (D-E) Effect of IpaC mutations on *S. flexneri* induction of actin polymerization-dependent membrane ruffles. HeLa cells infected with *S. flexneri* strains producing indicated IpaC protein at a MOI of 400. (D) Representative images. Red, polymerized actin; green, *S. flexneri*; blue, DNA. Scale bar, 10 μm. (E) Percentage of cells associated bacteria that are located in an actin polymerization-dependent membrane ruffle in experiments represented in panel D. (F-G) Efficiency of translocon pore insertion into plasma membranes of erythrocytes upon infection with *S. flexneri* strains producing indicated IpaC mutant at a MOI of 25. (F) Representative image of hemoglobin release. All wells are from the same experiment. (G) Hemoglobin release, quantified from three to five independent experiments represented in panel F. Control conditions, Media, WT, R362W, Δ*ipaC*, and SDS, are the same as Fig 4D; all mutants were tested in parallel. (H) Efficiency of translocon pore protein insertion into plasma membranes during infection of HeLa cells infected by *S. flexneri* strains producing indicated IpaC protein at a MOI of 500. Representative western blots of plasma membranes isolated by fractionation. IpaB and IpaC, translocon pore proteins; GroEL, bacterial cytoplasmic protein; caveolin-1, eukaryotic plasma membrane protein. (I-J) CytoD inhibits translocation by *S. flexneri* producing IpaC A354P. MEFs infected with *S. flexneri* strains at a MOI of 200. (I) Representative images; blue, DNA; red, mCherry (produced by all bacteria); green, TSAR. Scale bar 20 μm. (J) Number of bacteria with active secretion in experiments represented in panel I. (A-I) Data are representative of two to five independent experiments. Data points are independent experiments. Data are mean ± SEM. *, $p < 0.05$; **, $p < 0.01$; ***, $p < 0.001$ by one-way ANOVA with Dunnett's *post hoc* test comparing to WT (B, C, E, G) or Sidak *post hoc* test comparing the presence to the absence of cytoD (J).

Our data herein demonstrate that although docking is necessary for effector translocation [21], it is not sufficient. Our findings herein and our previous work together demonstrate that effector translocation depends on two distinct cellular processes: actin polymerization-induced conformational changes in the pore protein complex induces formation of translocon pores with open channels that enable activation of effector translocation (Fig 1), and intermediate filament binding to IpaC induces conformational changes in the pore protein complex that enable stable docking of bacteria onto the pore protein complex [21]. Because we found that in the absence of docking, pore opening occurs at wild-type levels (Fig 2B and 2C), yet in the absence of actin polymerization, docking occurs at reduced levels (Fig 1D), we favor a model in which actin polymerization induced conformational changes occur either prior to or simultaneously with intermediate filament interactions with IpaC (S8 Fig).

Our data demonstrate that bacterial docking is functionally separable from actin polymerization-dependent opening of the pore. In the absence of actin polymerization, the translocon pore proteins are sufficiently assembled to promote bacterial docking (Fig 1), a process that is known to depend on the T3SS translocon pore proteins [21,22]. Thus, whereas both actin polymerization and docking are necessary to trigger effector delivery, they are separable biological processes. These results indicate that the translocon pore proteins form a complex even in the absence of actin polymerization; as this complex does not have an open pore (Fig 2), it might appropriately be considered a closed intermediate pore complex.

Our isolation of an IpaC point mutant (A354P) that supports actin polymerization dependent pore opening but not membrane ruffle formation (Fig 5) demonstrates that these two actin dependent processes are independent. The observation that bacteria producing IpaC A354P translocate effectors but do not induce membrane ruffles is in line with the phenotypes of EHEC and EPEC, which require actin polymerization for translocation [25,26] but do not invade. Moreover, the *Yersinia* effector YopE represses pore formation by limiting actin polymerization [47]. These data suggest that the actin dependent opening of the pore channel is likely conserved among pathogens requiring T3SS activity whether or not they invade mammalian cells.

The observation that BCECF release is inhibited by cytoD, which blocks actin polymerization (Fig 2) demonstrates that actin polymerization promotes opening of the pore. Interestingly, bacteria producing IpaC A354P form pores in erythrocytes similarly to bacteria

producing WT IpaC but, as measured by TSAR activation, are less efficient at effector translocation (Fig 5). Why translocation occurs at lower levels is unclear. It is possible that the typical actin ruffles that occur with strains producing WT IpaC create membrane contact sites with the bacteria that promote effector translocation and that for strains producing IpaC A354P, because actin ruffles are diminished, fewer contact sites are formed. These additional membrane contacts may also enhance docking, as we observe a 2-fold reduction of docking in the presence of cytoD (Fig 1) and cytoD inhibits both actin-dependent pore opening and actin-dependent ruffles. In *Yersinia*, T3SS apparati are upregulated at sites of active secretion [48]; whether this also occurs for *S. flexneri* is not clear. An alternative explanation for the observed lower levels of translocation supported by IpaC A354P is that the character of the open pore is different; the presence of a proline in this position might lead to a steric effect that limits the efficiency of translocation. It is also possible that the proline mutation allows the opening of the translocon pore but limits the generation of a signal that is required to activate and/or regulate the magnitude of translocation. If a signal is required for translocation to occur, our investigation into the accessibility of the IpaC cysteine substitutions may provide relevant insights. We would anticipate that residues at the N-terminus, such as S17C, would label efficiently in the presence or absence of CytoD, as we predict they are located on the surface of the cell. Yet, we observed reduced labeling for S17C in the presence of cytoD (Fig 3), indicating that actin polymerization is necessary for this residue to be more accessible at the cell surface. We speculate that prior to opening of the pore, the N-terminal region of IpaC interacts more tightly with the T3SS needle. Such a stabilizing interaction could be beneficial to the pathogen: while the pore is assembling in the plasma membrane, this interaction could enable the delivery of additional pore proteins in the proximity of pore proteins that have already been delivered, and the release of the N-terminus from the needle could contribute to the generation of a signal to activate secretion. It is also possible that rather than actin polymerization *per se*, the presence of an intact actin cytoskeleton is necessary to promote pore opening and translocation. Further investigation into the nature of the interaction of the pore proteins with the T3SS needle will likely provide additional insights into the processes that promote translocation.

A major outstanding question in the field of type 3 secretion is how host cell contact is sensed and translated to activate effector secretion. Our data indicate that conformational or organizational changes to the translocon pore that arise from actin polymerization (this report) and interaction of the pore with intermediate filaments [24] are both required to activate secretion. Therefore, interaction of IpaC with intermediate filaments and/or actin polymerization trigger a signal to the T3SS sorting platform to initiate secretion through the T3SS of effectors. These findings are consistent with previous reports showing conformational changes to the pore were required to activate translocation by *Pseudomonas aeruginosa* [23]. Our documentation that actin polymerization induces conformational changes in the translocon pore provides support to the existing model that these conformational changes are transferred from the translocon pore through the needle [10,23,49,50] to the base of the T3SS [50] and then to the sorting complex, which is in the bacterial cytoplasm and triggers activation of secretion [51,52].

## Materials and methods

### Bacterial strains

For all experiments using *Shigella flexneri*, serovar 2a strain 2457T was used, and all strains are isogenic to it. *S. flexneri* was cultured in trypticase soy broth with appropriate antibiotics. Strains used in this study are listed in Table 1. The expression of recombinant IpaC was regulated by the pBAD promoter and induced by the inclusion of 1.2% arabinose in the media. *E.*

**Table 1. Strains and plasmids.**

| Strain | Source |
| --- | --- |
| *Shigella flexneri* strain 2457T | |
| *S. flexneri* 2457T Δ*ipaC* | Lab Stock |
| *S. flexneri* 2457T Δ*ipaC* pBAD33-WT IpaC | Lab Stock |
| *S. flexneri* 2457T Δ*ipaC* pBAD33-IpaC R362W | Lab Stock |
| *S. flexneri* 2457T Δ*ipaC* pBAD33-WT IpaC pTSAR | Lab Stock |
| *S. flexneri* 2457T Δ*ipaC* pBAD33-IpaC R362W pTSAR | Lab Stock |
| *S. flexneri* strain 2457T Δ*ipaC* pBAD33-WT IpaC pBR322-Afa-1 | Lab Stock |
| *S. flexneri* strain 2457T Δ*ipaC* pBAD33-IpaC R362W pBR322-Afa-1 | Lab Stock |
| *S. flexneri* 2457T pDSW206-OspB FLAG | Gift of Cammie Lesser |
| *S. flexneri* 2457T pDSW206-IpaA FLAG | Gift of Cammie Lesser |
| *S. flexneri* 2457T pDSW206-OspB TEM | Gift of Cammie Lesser |
| *S. flexneri* 2457T pTSAR | Lab stock |
| *E. coli* DH10B pSfT3SS | Gift of Cammie Lesser |
| *S. flexneri* strain 2457T Δ*ipaC* pBAD33-IpaC S17C pBR322-Afa-1 | Lab Stock |
| *S. flexneri* strain 2457T Δ*ipaC* pBAD33-IpaC A106C pBR322-Afa-1 | Lab Stock |
| *S. flexneri* strain 2457T Δ*ipaC* pBAD33-IpaC S349C pBR322-Afa-1 | Lab Stock |
| *S. flexneri* strain 2457T Δ*ipaC* pBAD33-IpaC A353C pBR322-Afa-1 | Lab Stock |
| *S. flexneri* strain 2457T Δ*ipaC* pBAD33-IpaC A358C pBR322-Afa-1 | Lab Stock |
| *S. flexneri* strain 2457T Δ*ipaC* pBAD33-IpaC S17C R362W pBR322-Afa-1 | Lab Stock |
| *S. flexneri* strain 2457T Δ*ipaC* pBAD33-IpaC A106C R362W pBR322-Afa-1 | Lab Stock |
| *S. flexneri* strain 2457T Δ*ipaC* pBAD33-IpaC S349C R362W pBR322-Afa-1 | Lab Stock |
| *S. flexneri* strain 2457T Δ*ipaC* pBAD33-IpaC A358C R362W pBR322-Afa-1 | Lab Stock |
| *S. flexneri* strain 2457T Δ*ipaC* pTSAR pBAD33-IpaC Q308A | This Study |
| *S. flexneri* strain 2457T Δ*ipaC* pTSAR pBAD33-IpaC Q312A | This Study |
| *S. flexneri* strain 2457T Δ*ipaC* pTSAR pBAD33-IpaC K316A | This Study |
| *S. flexneri* strain 2457T Δ*ipaC* pTSAR pBAD33-IpaC Q317A | This Study |
| *S. flexneri* strain 2457T Δ*ipaC* pTSAR pBAD33-IpaC Q323A | This Study |
| *S. flexneri* strain 2457T Δ*ipaC* pTSAR pBAD33-IpaC K326A | This Study |
| *S. flexneri* strain 2457T Δ*ipaC* pTSAR pBAD33-IpaC E327A | This Study |
| *S. flexneri* strain 2457T Δ*ipaC* pTSAR pBAD33-IpaC Q330A | This Study |
| *S. flexneri* strain 2457T Δ*ipaC* pTSAR pBAD33-IpaC Q334A | This Study |
| *S. flexneri* strain 2457T Δ*ipaC* pTSAR pBAD33-IpaC I336A | This Study |
| *S. flexneri* strains 2457T Δ*ipaC* pTSAR pBAD33-IpaC D344A | This Study |
| *S. flexneri* strain 2457T Δ*ipaC* pTSAR pBAD18-WT IpaC | This Study |
| *S. flexneri* strain 2457T Δ*ipaC* pTSAR pBAD18-IpaC I336P | This Study |
| *S. flexneri* strain 2457T Δ*ipaC* pTSAR pBAD18-IpaC L340P | This Study |
| *S. flexneri* strain 2457T Δ*ipaC* pTSAR pBAD18-IpaC Δcoiled-coil | This Study |
| *S. flexneri* strain 2457T Δ*ipaC* pBAD33-IpaC Q308P | This Study |
| *S. flexneri* strain 2457T Δ*ipaC* pBAD33-IpaC A354P | This Study |
| *S. flexneri* strain 2457T Δ*ipaC* pTSAR pBAD33-IpaC Q308P | This Study |
| *S. flexneri* strain 2457T Δ*ipaC* pTSAR pBAD33-IpaC A354P | This Study |
| *E. coli* DH10B | Invitrogen |
| **Plasmids** | Lab Stock |
| pBAD33-WT IpaC | Lab Stock |
| pBAD33-IpaC R362W | Lab Stock |
| pBAD33-IpaC S17C | Lab Stock |
| pBAD33-IpaC A106C | Lab Stock |

(*Continued*)

**Table 1.** (Continued)

| Strain | Source |
|---|---|
| pBAD33-IpaC S349C | Lab Stock |
| pBAD33-IpaC A353C | Lab Stock |
| pBAD33-IpaC A358C | Lab Stock |
| pBAD33-IpaC S17C R362W | Lab Stock |
| pBAD33-IpaC A106C R362W | Lab Stock |
| pBAD33-IpaC S349C R362W | Lab Stock |
| pBAD33-IpaC A358C R362W | Lab Stock |
| pBAD33-IpaC Q308A | This Study |
| pBAD33-IpaC Q312A | This Study |
| pBAD33-IpaC K316A | This Study |
| pBAD33-IpaC Q317A | This Study |
| pBAD33-IpaC Q323A | This Study |
| pBAD33-IpaC K326A | This Study |
| pBAD33-IpaC E327A | This Study |
| pBAD33-IpaC Q330A | This Study |
| pBAD33-IpaC Q334A | This Study |
| pBAD33-IpaC I336A | This Study |
| pBAD33-IpaC D344A | This Study |
| pBAD33-IpaC Q308P | This Study |
| pBAD33-IpaC A354P | This Study |
| pBAD18-WT IpaC | This Study |
| pBAD18-IpaC I336P | This Study |
| pBAD18-IpaC L340P | This Study |
| pBAD18-IpaC Δcoiled-coil | This Study |
| pTSAR | Gift of Claude Parsot |
| pBR322-Afa-1 | Gift of Cammie Lesser |
| pDSW206-OspB FLAG | Gift of Cammie Lesser |
| pDSW206-IpaA FLAG | Gift of Cammie Lesser |
| pDSW206-OspB TEM | Gift of Cammie Lesser |

*coli* pSfT3SS [39] was cultured in lysogeny broth (LB) containing appropriate antibiotics, and the expression of the T3SS in this strain was regulated by IPTG.

## Cell culture

HeLa cells were acquired from ATCC (CCL2). Vim$^{+/+}$ and Vim$^{-/-}$ MEFs were generously provided by Victor Faundez (Emory University). MEFs and HeLa cells were cultured in DMEM supplemented with 0.45% glucose and 10% FBS. All cells were cultured at 37˚C in humidified air containing 5% $CO_2$. All cells are periodically tested for mycoplasma.

## Bacterial effector translocation

For quantification of bacterial effector translocation into the cytosol of mammalian cells by western blot, HeLa cells were seeded at 3 x 10$^5$ cells per well in a six-well plate the day prior to the experiment. *S. flexneri* strains were cultured to exponential phase at 37˚C, and the expression of effectors was induced with 100 mM IPTG. HeLa cells were pretreated for 30 minutes prior to infection with or without cytochalasin D at 0.5 μg/mL. Bacteria were added to HeLa cells at a multiplicity of infection (MOI) of 200 and were centrifuged onto cells at 800 x *g* for

10 minutes at room temperature. Bacteria and HeLa cells were co-cultured for an additional 50 minutes at 37°C. The cells were washed with HBSS and lysed with RIPA buffer (50 mM Tris, pH 8 containing 150 mM NaCl, 1% Nonidet-P40, 0.1% SDS, 10 mM NaF, and EDTA-free protease inhibitor cocktail, Roche). Cellular debris and bacteria were removed by centrifugation and collected as the bacterial fraction. The abundance of OspB or IpaA delivered to the cytosol was determined by western blot.

## Translocation and docking

The measurement of docking and effector protein secretion was performed as previously described [21,24,40]. Briefly, HeLa cells or Vim[+/+] or Vim[-/-] MEFs were seeded onto a glass coverslip at 3 x 10⁵ cells per well of a six-well plate. *S. flexneri* that constitutively produce mCherry under the rpsM promoter and harbor the TSAR reporter were grown to exponential phase. GFP expression from the TSAR reporter is regulated by an MxiE dependent promoter, and MxiE transcription is induced by the secretion of OspD through the T3SS (S2A Fig). Cells were pretreated for 30 minutes prior to infection with or without cytochalasin D at 0.5 µg/mL or Latrunculin B (Sigma) at 0.5–10 µM. Bacteria were added to cells at an MOI of 200 and centrifuged onto the cells at 800 x *g* for 10 minutes at room temperature. The co-culture was incubated at 37°C for an additional 50 minutes. The infected cells were washed with HBSS and fixed with 3.7% paraformaldehyde. Coverslips were mounted onto glass slides with ProLong Diamond (Invitrogen). Bacteria were examined by epifluorescence microscopy. Bacterial docking was quantified by determining the number of mCherry-producing bacteria that remained associated with cells. Bacterial effector translocation was determined by counting the number of cell-associated bacteria expressing GFP.

## Congo red induced T3SS secretion

WT *S. flexneri* were grown to exponential phase, were recovered by centrifugation, and were resuspended in phosphate-buffered saline (PBS) with or without 10 µM Congo red in the presence or absence of cytoD at 0.5 µg/mL. The mixture was incubated for 60 minutes at 37°C. The bacteria were centrifuged at 15,000 x *g*, and the supernatant and pellet were collected and resuspended in equal volumes. Silver staining of SDS-PAGE gels was performed using Silver Stain Plus Kit (Bio-Rad). Alternatively, western blots were performed to assess the impact of cytoD and Congo red on the secretion of specific proteins.

## Measurement of actin tail formation

HeLa cells were seeded onto glass coverslips in the well of a six-well plate at 4 x 10⁵ cells per well. The following day the cells were infected with *S. flexneri* at a MOI of 200 in DMEM supplemented with 10% FBS. The bacteria were centrifuged onto the cells at 800 x *g* for 10 minutes at room temperature. The co-culture was allowed to incubate for an additional 40 minutes at 37°C with 5% $CO_2$. The cells were washed three times and the media was changed to DMEM supplemented with 10% FBS and either DMSO, 0.5 µg/mL of cytoD, or 5 µM of Latrunculin B. The infection was allowed to incubate for an additional 45 minutes at 37°C with 5% $CO_2$. The cells were washed five times with HBSS and fixed with 4% FBS in PBS for 30 minutes. The membranes were permeabilized by 1% Triton X-100 in PBS for 30 minutes at room temperature. The DNA was stained with Hoechst and the actin was stained with Alexa Fluor Plus 750 conjugated to Phalloidin (Invitrogen). Coverslips were mounted onto glass slides with Pro-Long Diamond (Invitrogen) and images were collected by epifluorescence microscopy. Images displayed are maximum intensity projections of Z-stacks that underwent a Richardson-Lucy deconvolution.

## Pore formation by BCECF release

For the measurement of pore formation in nucleated cells, a 2'-7'-bis-(2-carboxyethyl)-5-(and-6)-carboxyfluorescein (BCECF) release assay was performed, as described previously [21]. Briefly, 2 x $10^4$ HeLa cells were seeded per well in a 96-well plate the day prior to the experiment. On the day of the experiment, the media was removed, replaced with HBSS containing BCECF-AM (BCECF, Invitrogen, B1170) and Hoechst (Invitrogen), and cells were incubated for 30 minutes at 37°C in humidified air containing 5% $CO_2$ in the presence or absence of cytochalasin D at 0.5 μg/mL. The cells were washed with HBSS and were infected with *E. coli* pSfT3SS at an MOI of 100 (or as otherwise indicated), with centrifugation of bacteria onto the cells at 800 x *g* for 10 minutes at room temperature. The co-culture was incubated at 37°C for 1 hour. The co-culture was then centrifuged at 100 x *g* for four minutes, the media replaced, and images of live cells were collected by epifluorescence microscopy.

## Pore formation by erythrocyte lysis

Defibrinated sheep erythrocytes (HemoStat) were pelleted at 2,000 x *g* and resuspended in 100 μL of PBS. Cells were infected at an MOI of 25 in 100 μl of PBS supplemented as appropriate with 1.2% arabinose to induce expression of bacterial virulence factors. Bacteria were centrifuged onto the erythrocytes at 2,000 x *g* for 10 minutes at 25°C and were co-cultured with the erythrocytes for 50 minutes at 37°C, after which the bacteria and erythrocytes were pelleted at 2,000 x *g* for 10 minutes at 25°C. As a positive control for lysis, a portion of uninfected erythrocytes were treated with 0.02% SDS. The supernatants were collected, and their absorbance at 570 nm was determined using an Epoch II plate reader (BioTech).

## Labeling of IpaC cysteine residues by PEG5000-maleimide

The labeling of cysteine substitutions in IpaC by PEG5000-maleimide was performed as previously described [24,40]. HeLa cells were seeded at 4 x $10^5$ cells per well in a six-well plate. HeLa cells were pretreated for 30 minutes prior to infection with or without cytochalasin D at 0.5 μg/mL. *S. flexneri* strains harboring the Afa-1 pilus and indicated IpaC mutant were cultured to exponential phase at 37°C, and production of IpaC was induced by addition of 1.2% arabinose for 2 hours. Cells were infected at an MOI of 500 in 50 mM Tris, pH 7.4, supplemented with 150 mM NaCl, 1.2% arabinose, and 2.5 mM PEG5000-maleimide, with or without cytoD at 0.5μg/mL. The bacteria were centrifuged onto the cells at 800 x *g* for 10 minutes at 25°C and were co-cultured with the cells for an additional 20 minutes at 37°C. The cells were washed with ice-cold 50 mM Tris, pH 7.4, supplemented with protease inhibitors, and the cells were scraped from the dishes. The membranes were isolated from the cells by detergent fractionation as previously described [21,24,40]. Cells were centrifuged at 3,000 x *g* for three minutes at 25°C, resuspended in 50 mM Tris, pH 7.4, supplemented with protease inhibitors and 0.2% saponin, and were incubated on ice for 20 minutes. The cells were pelleted at 21,000 x *g* for 30 minutes at 4°C; the resulting supernatant contained the cytosolic fraction. The pellet was resuspended in 50 mM Tris, pH 7.4, supplemented with protease inhibitors and 0.5% Triton X-100, was incubated for 30 minutes on ice, and was centrifuged at 21,000 x *g* for 15 minutes at 4°C. The resulting supernatant contained the solubilized membrane, and the resulting pellet contained bacteria, cellular nuclei, and debris. The efficiency of IpaC labeling by PEG-5000 maleimide was determined by western blot.

## Quantification of pore protein insertion into plasma membranes

HeLa cells were seeded at 4 x $10^5$ cells per well in a six-well plate. HeLa cells were pretreated for 30 minutes prior to infection with or without cytochalasin D at 0.5 μg/mL. *S. flexneri*

strains harboring indicated IpaC mutant were cultured to exponential phase at 37°C and, IpaC production was induced by addition of 1.2% arabinose for 2 hours. Cells were infected at an MOI of 500 in DMEM supplemented with 10% FBS with or without cytoD at 0.5μg/mL. The bacteria were centrifuged onto the cells at 800 x *g* for 10 minutes at 25°C, and were co-cultured with cells for an additional 20 minutes at 37°C. The cells were washed with ice-cold 50 mM Tris, pH 7.4, supplemented with protease inhibitors, and the cells were scraped from the dishes. The membranes were isolated from the cells by detergent fractionation as described above and previously described [21,24,40]. The efficiency of IpaC labeling by PEG-5000 maleimide was determined by western blot.

## Quantification of actin-mediated membrane ruffle formation

The day prior to infection, HeLa cells were seeded on coverslips at $4 \times 10^5$ cells per well in a six-well plate. Cells were infected at an MOI of 400 and were centrifuged onto the cells at 800 x *g* for 10 minutes at room temperature. The cells and bacteria were co-cultured for 30 minutes at 37°C. The cells were washed five times with warm HBSS and fixed with 3.7% PFA in F buffer for 20 minutes. The cells were permeabilized for 30 minutes with 1% Triton X-100 at room temperature on a rocker at low speed, were washed with PBS, and were blocked with 10% goat serum in PBS for 30 minutes. The cells were then incubated overnight at 4°C with rabbit anti-*Shigella* conjugated to FITC (ViroStat, catalog no. 0903). The next day, the cells were washed with PBS and were incubated with 1:50 dilution of phalloidin-Alexa Fluor 594 (ThermoFisher, Cat# A12381) for 30 minutes at room temperature. After additional washing, and the coverslips were mounted onto slides with Prolong Diamond (ThermoFisher, cat # P36970). The efficiency of ruffle formation was determined as the percentage of cell-associated bacteria with polymerized actin outlining the bacteria.

## Generation of IpaC mutants by splice-overlap-PCR

Charged residues within the coiled-coil region were replaced with alanine by splice overlap PCR mutagenesis using Accuprime *pfx* polymerase (Invitrogen). PCR products containing the alanine mutation were cloned under the control of the *ara* promoter by insertion into pBAD33 by digestion with Kpn1 (NEB) and Sph1 (NEB). The plasmids were expressed in *S. flexneri* 2457T Δ*ipaC* pTSAR. The same approach was used to generate IpaC lacking the coiled-coil region, residues 308–344.

## Screen to identify IpaC residues required for actin polymerization-dependent translocation

A library of IpaC mutants with missense mutations in the coiled-coil domain and flanking regions was generated using error-prone PCR with the GeneMorphII (Agilent) domain mutagenesis kit. The library was cloned under the control of the *ara* promoter and was expressed in *S. flexneri* 2457T Δ*ipaC* pTSAR. The resulting strains were arrayed and were used to infect MEFs seeded at $2 \times 10^4$ cells per well in a 96-well plate. The bacteria were pelleted onto the cells at 800 x *g* for 10 minutes at 25°C. Following an additional 50 min incubation at 37°C, the cells were washed and fixed with 3.7% paraformaldehyde. Fixed cells were stained with Hoechst and were imaged using a Cell Discover 7 automated microscope at the Harvard Center for Biological Imaging. Images were manually screened to identify IpaC variants that supported fewer GFP-positive bacteria associating with cells. 137 clones meeting these criteria were identified; for each, the *ipaC*-containing plasmid DNA was isolated, and *ipaC* was sequenced. 131 of the *ipaC* mutants contained non-sense or frameshift mutations. Six contained one or two missense mutations.

## Copper mediated crosslinking of IpaC

HeLa cells were seeded at $4 \times 10^5$ cells per well in a 6-well plate. The cells were washed once with Hank's Balanced Salt Solution (HBSS) containing 4% FBS and 1.2% arabinose. The cells were infected at an MOI of 200 in HBSS containing 4% FBS and 1.2% arabinose, with or without 25 μM copper phenanthroline [23]. The bacteria were centrifuged onto cells at 800 *g* for 10 minutes at 25˚C and incubated at 37˚C in humidified air with 5% $CO_2$ for 10 min. IpaC delivered to cell membranes was recovered as done previously [23]. Briefly, cells were washed with HBSS and lysed with 0.5% Triton X-100, and bacteria and cellular debris were removed by two successive centrifugations at 21,000 *g* for 2 minutes each at 25˚C.

## Antibodies for western blots

The following antibodies were used for western blots: mouse anti-FLAG, (Sigma, catalog no. F1804) (1:10,000), rabbit anti-TEM β-lactamase (5 Prime-3 Prime Inc., catalog no. 7–661211) (1:1,000), rabbit anti-β-actin conjugated with HRP (Sigma, catalog no. A3854) (1:20,000), rabbit anti-IpaC (gift from Wendy Picking; diluted 1:10,000), mouse anti-IpaB clone 1H4 (Gift of Robert Kaminski; diluted 1:10,000); rabbit anti-GroEL (Sigma, catalog no. G6352) (1:1,000,000), rabbit anti-caveolin-1 (Sigma, catalog no. C4490), rabbit anti-SepA [53] (1:1000), mouse anti-IpgD (gift of Armelle Phalipon [54]) (1:3000), goat anti-rabbit conjugated with horseradish peroxidase (HRP) (Jackson ImmunoResearch, catalog no. 115-035-003) (1:5,000), goat anti-mouse conjugated with HRP (Jackson ImmunoResearch, catalog no. 111-035-003) (1:5,000).

## Microscopy and image analysis

Images were collected using a Nikon TE-300 or Nikon TE-2000S microscope equipped with Q-Imaging Exi Blue Cameras (Q-imaging), Chroma Filters, and IVision Software (BioVision Technologies), or a Nikon Ti-2 microscope equipped with an Iris15 camera (Photometrics) and an Orca Fusion-BT camera (Hamamatsu), Semrock filters, and NIS-Elements software (Nikon). Unless noted otherwise, images were randomly collected across a coverslip. Single channel images were pseudo-colored and assembled in Photoshop (Adobe).

Chemiluminescent signals from western blots were captured by film. The developed film was scanned using a Perfection 4990 scanner (Epson), and the density of bands was determined using ImageJ (NIH).

## Statistical analysis

For the comparison of means, a Student's t-test was used to compare experiments containing two groups and an ANOVA was used for experiments containing three or more groups using Prism 9 (GraphPad Software). Unless otherwise noted, at least three independent experiments were performed on independent days using independent bacterial cultures. Data points represent independent experiments.

## Supporting information

**S1 Fig. T3SS effector translocation requires actin polymerization.** (A-D) *S. flexneri* translocation of the FLAG-tagged type 3 effector IpaA and the TEM β-lactamase-tagged type 3 effector OspB into HeLa cells requires actin polymerization. (A and C) Representative western blots of cytosolic FLAG-tagged IpaA (A) or TEM-tagged OspB (C) in *S. flexneri* infected HeLa cells at a MOI of 200. GroEL, bacterial cytosolic protein; actin, eukaryotic cytosolic protein. Each lane is an independent well from one experiment. (B and D) Quantification of cytosolic

effectors FLAG-tagged IpaA (B) and TEM-tagged OspB (D) from experiments depicted in panels A and C, respectively. *, p<0.05; ***, p<0.001; Student's t-test. (E) Schematic diagram showing differences between induced type 3 mediated secretion from bacteria in liquid media, which results in bacterial effectors in the extracellular medium, and plasma membrane contact-induced type 3 mediated translocation, which results in bacterial effector protein translocation into the host cytosol. (F-G) Effect of CytoD on type 3 secretion of effectors following induction with Congo red. (F) Bacterial supernatant proteins detected in silver-stained gel. Blots are representative of four experiments; two biological replicates were performed in each experiment. (G) Western blots of bacterial supernatants and pellets, representative of three independent experiments. (F-G) IpaA, IpaB, IpaC, IpaD, IpaH's, and IpgD, type 3 secreted proteins. GroEL, bacterial cytoplasmic protein; SepA and IcsA, type 5 secreted proteins, whose secretion occurs independent of both the T3SS and Congo red.
(TIF)

**S2 Fig. Type 3 secretion activity, as measured by TSAR, requires actin polymerization.** (A) Schematic depiction of OspD-dependent production of GFP by the TSAR reporter. Transient contact with host plasma membrane activates secretion of the translocon pore proteins IpaC and IpaB (IpaB not shown) (I). The secretion of IpaC and IpaB liberates their cognate chaperone, IpgC, and secreted IpaB and IpaC form the translocon pore in the plasma membrane, onto which the bacterium docks (II). OspD translocation liberates its chaperone, MxiE. IpgC binds to and activates MxiE, and IpgC-MxiE functions as a transcriptional activator, inducing the *mxiE* promoter upstream of *gfp* (III). HeLa cells infected with *S. flexneri* carrying TSAR with or without cytoD (B-C) or latrunculin B (D-E) at a MOI of 200. (B and D) Representative fluorescent images of cells treated with or without 5 μm Lat. B. Blue, DNA (Hoechst); red, mCherry (constitutively produced); green, GFP (transcriptionally activated by the secretion of OspD). Scale bar 10 μm (B) or 20 μm (D). (C and E) Percentage of docked bacteria with active secretion in experiments represented in panel B or D. Data points represent independent experiments. *, p<0.05; Student's t-test.
(TIF)

**S3 Fig. Actin tail formation is blocked by cytochalasin D or latrunculin B.** Actin tail formation in HeLa cells infected with *S. flexneri* at an MOI of 200 in the presence or absence of cytochalasin D or latrunculin B. (A) Representative fluorescent microscopy images. Blue, DNA; green, bacteria; red, actin. Scale bar 20 μm. (B) Quantification of the efficiency of actin tail formation from images presented in panel A. Data are mean ± SEM of three independent experiments. Data points are independent experiments. *, p<0.05, one-way ANOVA with Dunnett's *post hoc* test.
(TIF)

**S4 Fig. Formation of intermediary pores in the plasma membrane requires actin polymerization.** BCECF dye released from HeLa cells infected with *E. coli* pSfT3SS as a function of multiplicity of infection. Data are mean ± SEM of two independent experiments; data points are independent experiments.
(TIF)

**S5 Fig. Copper oxidant mediated crosslinking of IpaC requires actin polymerization.** (A) Schematic depiction of the position of S17C and A353C in IpaC indicated by asterisk. (B) Schematic depiction showing that disulfide bonds do not form between adjacent IpaC monomers when A353C is in the cytosol (I), but can form in an intermediary pore complex when an A353C-containing loop of IpaC extends into the lumen of the pore (II). (C-D) Effect of cytoD on the ability of the oxidant copper to induce crosslinking between IpaC monomers at S17C

and A353C. HeLa cells were infected at a MOI of 200. (C) Representative western blot. (D) Quantification of crosslinked dimer band density in panel D. Data are the mean ± SEM of three independent experiments. Data points are independent experiments. ***, p<0.001 by two-way ANOVA with Sidak *post hoc* test.
(TIF)

**S6 Fig. Scanning alanine mutagenesis of charged/polar residues in the coiled-coil domain did not identify individual residues critical for docking or effector translocation.** (A) Prediction of coiled-coil domains in IpaC by COILS [55]; 14, 21, and 28 indicate the number of amino acids in each coil. (B) Western blot of IpaC produced by *S. flexneri* Δ*ipaC* strains induced to produce IpaC alleles or GroEL, a cytoplasmic bacterial protein. (C-E) Docking and translocation into MEFs infected at a MOI of 200 by *S. flexneri* strains producing indicated IpaC alanine mutant. (C) Docked bacteria per cell at 50 minutes of infection. (D) Number of bacteria with active secretion per cell. (E) Percentage of docked bacteria with active secretion. (C-E) Data are mean ± SEM from two to three independent experiments; data points represent individual experiments.
(TIF)

**S7 Fig. Quantification of IpaC and IpaB in the plasma membrane.** Quantification of western bands of IpaC (A) or IpaB (B) from Fig 5H. Data are mean ± SEM from three independent experiments. *, P<0.05; one-way ANOVA with Dunnett's *post hoc* test.
(TIF)

**S8 Fig. Model of actin polymerization induced opening of the translocon pore.** Contact of the T3SS with the host plasma membrane (I) induces the T3SS to deliver the translocon pore proteins into the plasma membrane (II). Actin polymerization opens the pore and the interaction of IpaC with intermediate filaments promotes bacterial docking onto the pore complex (III). Effectors are secreted through the T3SS, and together with IpaC, trigger membrane ruffle formation (IV) and consequent bacterial uptake.
(TIF)

**S1 Data. Data contained in graphs.**
(XLSX)

## Acknowledgments

We thank Claude Parsot, Wendy Picking, Annalise Reeves and Cammie Lesser, Robert Kaminski, and Armelle Phalipon for reagents. We thank members of the Goldberg laboratory, Cammie Lesser, and Amy Barczak for helpful discussions. We thank Douglas Richardson and the Harvard Center for Biological Imaging for infrastructure and support.

## Author Contributions

**Conceptualization:** Brian C. Russo, Jeffrey K. Duncan-Lowey, Marcia B. Goldberg.

**Data curation:** Brian C. Russo, Jeffrey K. Duncan-Lowey.

**Formal analysis:** Brian C. Russo, Jeffrey K. Duncan-Lowey, Poyin Chen, Marcia B. Goldberg.

**Funding acquisition:** Brian C. Russo, Poyin Chen, Marcia B. Goldberg.

**Investigation:** Brian C. Russo, Jeffrey K. Duncan-Lowey, Poyin Chen.

**Project administration:** Brian C. Russo, Marcia B. Goldberg.

**Supervision:** Marcia B. Goldberg.

**Writing – original draft:** Brian C. Russo.

**Writing – review & editing:** Brian C. Russo, Jeffrey K. Duncan-Lowey, Poyin Chen, Marcia B. Goldberg.

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
