## [Decision Letter · Decision Letter 0]

31 Jul 2021

Dear Dr. Goldberg,

Thank you very much for submitting your manuscript "The type 3 secretion system requires actin polymerization to open translocon pores" for consideration at PLOS Pathogens. As with all papers reviewed by the journal, your manuscript was reviewed by members of the editorial board and by several independent reviewers. In light of the reviews (below this email), we would like to invite the resubmission of a significantly-revised version that takes into account the reviewers' comments.

The reviewers agree on the merits of the manuscript, but found issues with the discussion being one-sided and lacking alternative interpretations. This is important since the majority of the conclusions is driven by cytochalasin treatment of cells, which may lead to both direct and indirect alterations in the host cells. These different effects can alter interpretation of the data presented and should be discussed in a more balanced manner.

Reviewer #1 also recommends some additional experiments. These suggested experiments can be challenging as noted by the reviewer so they are not necessary. However, if any can be performed, the results would certainly contribute to the manuscript. One other possibility that might be more feasible is to wash out the cytochalasin and monitoring cells that regain the ability to polymerize actin are the same cells associated with Shigella that regain the ability to actively injecting OspD. Again, this is not necessary, but would contribute to the interpretation of the data.

Minor comments

Legends should include MOI

We cannot make any decision about publication until we have seen the revised manuscript and your response to the reviewers' comments. Your revised manuscript is also likely to be sent to reviewers for further evaluation.

Sincerely,

Vincent T Lee

Associate Editor

PLOS Pathogens

Guy Tran Van Nhieu

Section Editor

PLOS Pathogens

Kasturi Haldar

Editor-in-Chief

PLOS Pathogens

orcid.org/0000-0001-5065-158X

Michael Malim

Editor-in-Chief

PLOS Pathogens

orcid.org/0000-0002-7699-2064

The reviewers agree on the merits of the manuscript, but found issues with the discussion being one-sided and lacking alternative interpretations. This is important since the majority of the conclusions is driven by cytochalasin treatment of cells, which may lead to both direct and indirect alterations in the host cells. These different effects can alter interpretation of the data presented and should be discussed in a more balanced manner.

Reviewer #1 also recommends some additional experiments. These suggested experiments can be challenging as noted by the reviewer so they are not necessary. However, if any can be performed, the results would certainly contribute to the manuscript. One other possibility that might be more feasible is to wash out the cytochalasin and monitoring cells that regain the ability to polymerize actin are the same cells associated with Shigella that regain the ability to actively injecting OspD. Again, this is not necessary, but would contribute to the interpretation of the data.

Minor comments

Legends should include MOI

Reviewer's Responses to Questions

**Part I - Summary**

Reviewer #1: Russo et al. provide interesting results supporting the idea that host actin polymerization may induce conformational changes in the Shigella translocation pore protein IpaC, thereby opening pores that allow translocation of effector proteins into host cells. Together with previous work on the role of host vimentin proteins, this study suggests that Shigella may manipulate the functions of two different host cytoskeletal elements to promote distinct and separable steps in effector protein translocation. The data in this manuscript suggests that actin polymerization controls the opening of translocon pores, and previously published results show that binding of IpaC to vimentin induces conformational changes in the pore complex that allow stable docking of Shigella.

This is an interesting study that is generally well performed. My main substantive criticism is that the evidence that host actin polymerization is needed to open translocon pores is somewhat weak, in that it depends entirely on a single approach (i.e. cytochalasin D treatment). The effects of cytochalasin D treatment are not confirmed through the use of other methods to inhibit actin polymerization (e.g. treatment with latrunculin A or RNAi-mediated depletion of components of the Arp2/3 complex or other host proteins that promote actin filament assembly). Moreover (and most importantly), the actin polymerization event that is inferred to happen (based on experiments with cytochalasin D) is never directly shown to occur during docking or effector translocation through the use of microscopy-based approaches. As a result, while the data certainly suggest a role for actin polymerization in opening of translocon pores, the work falls short of unequivocally demonstrating that Shigella stimulates actin polymerization to promote effector protein translocation. Perhaps only an intact host actin cytoskeleton (which would be disrupted by cytochalasin D treatment) is needed for translocation, without bacteria directly and locally stimulating actin filament assembly.

Reviewer #2: The findings presented here show that there is an interplay between the host actin cytoskeleton and the Ipa proteins that make up the membrane-embedded translocon pore of the Shigella T3SS. Specific signals elicited by IpaC promote actin polymerization which opens the translocon pore to permit the subsequent passage of effector proteins into the host cytosol. The authors have already attributed a role for intermediate filaments in the docking of the T3SS needle and translocon to the host cell with IpaC also contributing to this process, however, the actin polymerization involved with translocon opening is distinct from that process. Thus, these findings are new and propel the field forward with regard to understanding the molecular events controlling translocation. The manuscript also puts this process into the context of what is known for other T3SS. The experiments described are elegantly designed and the authors are to be credited with the fact that these are not particularly simple experiments to execute. I will also upload my separately written review.

Reviewer #3: NoneEffector secretion by T3SS is triggered by host cell contact, the mechanism by which host cell contact is sensed is unclear but involves the translocon. Russo et al. present evidence that actin polymerization in the host cell is needed for effector secretion to commence. This involves a conformational change in the translocation pore and is distinct from the intermediate filament-mediated docking of the needle-tip to the translocation pore. The experiments are well executed and the data is presented clearly. My primary comments regard the interpretation of the results. Specifically, does actin polymerization really mediate “pore opening” as opposed to a conformational change that “flips the switch” to effector export in the bacterial cytoplasm (presumably by propagating the conformational change from the pore down the needle to the base of the apparatus)?

**Part II – Major Issues: Key Experiments Required for Acceptance**

Reviewer #1: My main substantive criticism is that the evidence that host actin polymerization is needed to open translocon pores is somewhat weak, in that it depends entirely on a single approach (i.e. cytochalasin D treatment).

In this regard, the work would be strengthened by the use of other approaches to inhibit actin polymerization (e.g. treatment with latrunculin A and/or RNAi-mediated depletion of components of the Arp2/3 complex or other host proteins that promote actin filament assembly).

The work would be strengthened even more by data that more directly demonstrates that Shigella locally stimulates actin polymerization during docking and/or effector translocation through the use of microscopy-based approaches. However, I recognize that obtaining such data might be quite difficult if these localized actin polymerization events are weak, highly transient, and/or masked by actin polymerization in ruffles. If experiments to directly detect actin polymerization during docking/translocation cannot be performed because of these technical issues, it would be useful for the authors to include a statement in the discussion section that explains the extent to which their data supports the idea that Shigella induces host actin polymerization to promote opening of translocon pores versus the possibility that pore opening instead simply requires an intact actin cytoskeleton that is not actively manipulated by Shigella.

Reviewer #2: No additional experiments are needed.

Reviewer #3: none

**Part III – Minor Issues: Editorial and Data Presentation Modifications**

Reviewer #1: • Abstract: ‘An IpaC mutant is identified that shows…’. It seems incorrect to state that an IpaC mutant ‘shows’, since it is really the results obtained with this mutant that show something. I would be inclined to change this sentence to something like ‘Results from experiments involving an IpaC mutant defective in … show that . ‘

• Page 4, first paragraph, lines 48-51. References are needed to support these statements.

• Page 6 and Fig. 1. ‘Actin polymerization is required for type 3 effector protein translocation. What is the evidence that cytochalasin D treatment inhibits actin polymerization under the conditions used in these experiments? Are the levels and/or distribution of F-actin altered by cytochalasin D treatment in these studies? Are similar results obtained by other inhibitors of actin polymerization such as latrunculin A or by genetic inhibition of actin polymerization (e.g. siRNA-mediated depletion of components of the Arp2/3 complex)?

• Fig. 1A. IpaC should be labeled in this diagram. On page 5, it is stated that two proteins comprise the translocation pore in bacterial type III secretion systems. What protein apart from IpaC comprises the translocation pore for Shigella? Is this other protein depicted in Figure 1AI, or is only IpaC shown?

• Figure 1 parts E and F. ANOVA and a post-test should be used for statistical analysis. T-tests should be used only for analysis of two data sets (Olsen, 2014. Infect. Immun. 82: 916-020.

• Figure S1, parts F and G. Duplicate samples seem to be loaded on these gels, and it would be good to explicitly state that these are technical replicates.

• Figure 2E. Why were IpaB levels in the membrane not quantified?

• Methods. The correct spelling is ‘scraped’, and not ‘scrapped’.

• Line 158. Maleimide is misspelled.

• Legend to Figure 3 line 705. Should ‘A353’ really be ‘A358’?

• Figure 3A,C,D. It seems strange that the PEG-maleimide accessibility of A358, which is depicted as inside the host cytosol (or close to it) is not affected by interaction with vimentins. Isn’t the IpaCR362W mutant supposed to be defective in activating the translocon pore? If so, then how can PEG-maleimide get into the host cytosol to label A358 in IpaC?

• Legend to Figure 3D. Some of the data are apparently from only two experiments. However, statistical analysis cannot (or at least should not) be performed on only two data sets. In my view, it is essential to perform all experiments at least three times and perform statistical analysis on the resulting three data sets to confirm significance and reproducibility.

• Since IpaC is known to contribute to actin polymerization, could the lack of effect of cytochalasin D treatment on PEG-maleimide labeling of IpaC.S17C be due to a defect in the ability of IpaC to promote actin polymerization? In this case, I’m not referring to actin polymerization in ruffles, but rather at sites of translocon opening.

• Line 202 and legend to Figure S4. Is ‘A353C’ correct, or do the authors mean ‘A358C’?

• Figure S4. It would be nice to have a control involving a cysteine mutation around the N-terminus of IpaC. If I am understanding this approach correctly, then such a mutant protein would be cross-linked in a manner that is unaffected by cytochalasin D treatment, since the cysteine residue would not be in the cytosol of the host cell.

• Figure 4. Are the IpaC coiled coil domain mutants expressed at levels similar to that of wild-type IpaC?

• Page 14, lines 268-269. The reader should be referred to Figure 5F,G.

• Figure 5H. Quantified Western blotting data from three experiments and resulting statistical analysis should be performed. By eye, IpaB and IpaC translocation seem to be slightly reduced for the Q308P mutant.

Reviewer #2: A couple of minor comments are included in the uploaded review.

Reviewer #3: BCECF release. I always thought that the actin polymerization in the absence of effector translocation dissociated the pore from the needle, allowing efflux. It’s less of an “open pore” and more of a dissociation of the pore from the needle assay then? (e.g. Viboud and Bliska 2001 EMBO J)

In this context, in Fig. 3, why is labeling in the presence of cytoD generally much weaker than without, even for extracellular residues, such as S17C. If BCECF release is due to actin polymerization-dependent dissociation of the pore from the needle-tip, could the lack of accessibility in the presence of cytoD mean that the reagent cannot access the cysteines in the assembled needle-tip/pore complex? The same could be going on with the reduced disulfide bond formation at position S17 in the presence of copper (Fig. S4C/D).

Notably, to argue against my own hypothesis. The reduction in docking upon cytoD treatment (Fig. 1E) would argue against stabilization of the translocon. However, in general, I find that the data could be argued both ways, and perhaps the authors could spend a bit more time on alternative explanations for their data? To my mind, the question of whether triggering of effector secretion is the result of opening of the translocation pore, or a conformational change in the translocation pore that results in a change in export specificity is not resolved.

Another point that the authors could mention is that the notion of a conformational change in the translocation pore triggers effector secretion has also been found in Pseudomonas aeruginosa (Armentrout 2016 PLoS Pathogens). Notably, the minor translocator, PopD, which is analogous to IpaC in that system, was implicated in the signal transduction process. It is interesting that, while these systems are different (PopD topology differs from IpaC, cytoD does not interefere with translocation, etc), the overall mechanism for engaging the T3SS seems to be conserved (at least the 10000ft view of it).

Fig. 5D-E – If I read the literature correctly, then membrane ruffling is promoted not only by IpaC, but also by effector proteins that are injected into the host cell. As a result, you don’t really know if these mutants are defective in IpaC-mediated ruffle-formation, since the phenotype could be indirect, a reflection of the defect in effector injection. Hemolysis, likely a direct function of the translocation pore, is less severely affected by the mutations than activation of the TSAR reporter or membrane ruffling, which requires the additional step of activating effector injection (and efficient effector delivery). Which means multiple steps could be compromised here. In general, I feel that the data in Fig. 5 could benefit from a more nuanced analysis of the potential pitfalls in their interpretation.

PLOS authors have the option to publish the peer review history of their article (what does this mean?). If published, this will include your full peer review and any attached files.

Reviewer #1: No

Reviewer #2: No

Reviewer #3: No
---

## [Decision Letter · Decision Letter 1]

31 Aug 2021

Dear Dr. Goldberg,

We are pleased to inform you that your manuscript 'The type 3 secretion system requires actin polymerization to open translocon pores' has been provisionally accepted for publication in PLOS Pathogens.

Best regards,

Vincent T Lee

Associate Editor

PLOS Pathogens

Guy Tran Van Nhieu

Section Editor

PLOS Pathogens

Kasturi Haldar

Editor-in-Chief

PLOS Pathogens

orcid.org/0000-0001-5065-158X

Michael Malim

Editor-in-Chief

PLOS Pathogens

orcid.org/0000-0002-7699-2064

Thank you for addressing the reviewers comments in a positive and constructive manner. Congratulations!

Reviewer Comments (if any, and for reference):

Reviewer's Responses to Questions

**Part I - Summary**

Reviewer #1: (No Response)

**Part II – Major Issues: Key Experiments Required for Acceptance**

Reviewer #1: I thank the authors for satisfactorily addressing my comments and queries.

**Part III – Minor Issues: Editorial and Data Presentation Modifications**

Reviewer #1: (No Response)

PLOS authors have the option to publish the peer review history of their article (what does this mean?). If published, this will include your full peer review and any attached files.

Reviewer #1: No

---

## [Editor Report · Acceptance letter]

6 Sep 2021

Dear Dr. Goldberg,

We are delighted to inform you that your manuscript, "The type 3 secretion system requires actin polymerization to open translocon pores," has been formally accepted for publication in PLOS Pathogens.

Best regards,

Kasturi Haldar

Editor-in-Chief

PLOS Pathogens

orcid.org/0000-0001-5065-158X

Michael Malim

Editor-in-Chief

PLOS Pathogens

orcid.org/0000-0002-7699-2064